# Natural Flavonol, Myricetin, Enhances the Function and Survival of Cryopreserved Hepatocytes In Vitro and In Vivo

**DOI:** 10.3390/ijms20246123

**Published:** 2019-12-04

**Authors:** Changhao Cui, Shin Enosawa, Hitomi Matsunari, Hiroshi Nagashima, Akihiro Umezawa

**Affiliations:** 1Division for Advanced Medical Sciences, National Center for Child Health and Development, Tokyo 157-8535, Japan; changhaocui@dlut.edu.cn; 2Laboratory of Developmental Engineering, School of Agriculture, Meiji University, Kanagawa 214-8571, Japan; hitomim@meiji.ac.jp (H.M.); hnagas@meiji.ac.jp (H.N.); 3Center for Regenerative Medicine, National Center for Child Health and Development, Tokyo 157-8535, Japan; umezawa-a@ncchd.go.jp

**Keywords:** myricetin, hepatocyte, culture, cell transplantation, ornithine transcarbamylase, kusabira orange

## Abstract

To improve the therapeutic potential of hepatocyte transplantation, the effects of the mitogen-activated protein kinase kinase 4 (MKK4) inhibitor, myricetin (3,3′,4′,5,5′,7-hexahydroxylflavone) were examined using porcine and human hepatocytes in vitro and in vivo. Hepatocytes were cultured, showing the typical morphology of hepatic parenchymal cell under 1–10 µmol/L of myricetin, keeping hepatocyte specific gene expression, and ammonia removal activity. After injecting the hepatocytes into neonatal Severe combined immunodeficiency (SCID) mouse livers, cell colony formation was found at 10–15 weeks after transplantation. The human albumin levels in the sera of engrafted mice were significantly higher in the recipients of myricetin-treated cells than non-treated cells, corresponding to the size of the colonies. In terms of therapeutic efficacy, the injection of myricetin-treated hepatocytes significantly prolonged the survival of ornithine transcarbamylase-deficient SCID mice from 32 days (non-transplant control) to 54 days. Biochemically, the phosphorylation of MKK4 was inhibited in the myricetin-treated hepatocytes. These findings suggest that myricetin has a potentially therapeutic benefit that regulates hepatocyte function and survival, thereby treating liver failure.

## 1. Introduction

Orthotopic liver transplantation has been successful in the treatment of a variety of end-stage liver diseases [1,2]. The conditions under which patients require liver transplantation involve diseases that range from acute to chronic liver failure. The former is generally characterized by a rapid deterioration in total liver function, resulting in coagulopathy and encephalopathy; the latter develops slowly but irreversibly as seen with cholestatic liver disease, cirrhosis, and congenital metabolic disorders. Although liver transplantation achieves good results, the procedure, i.e., the replacement of a large organ is a heavy burden on patients and medical staff. Cell transplantation has the potential to be an alternative and more accessible treatment. Thus far, over 100 cases of clinical hepatocyte transplantation have been reported [3,4]. Although there are currently not enough results from which to draw definitive conclusions, there has been an increase in research focusing on congenital metabolic diseases, particularly in the field of pediatrics [5]. Among these, urea cycle disorders are thought to be one of the most suitable indications for hepatocyte transplantation in clinical trials [6,7,8]. There are two typical disorders, ornithine transcarbamylase (OTC)-deficiency and carbamoylphosphate synthetase 1-deficiency, both of which arise from congenital genetic mutations. Depending on the level of remaining normal liver function, patients can present with fulminant or chronic disorders. Severe disorders are often associated with infantile male OTC-deficiency because the *OTC* gene is located on the X chromosome. Common treatments include a non- or low-protein diet, medicating with benzoic acid, phenylbutyrate (Buphenyl^R^), arginine, and citrulline, and hemodialysis as a backstop for a hyperammonemia attack. Although liver transplantation is an efficient and permanent treatment option, it is associated with substantial risk in infantile surgery. As such, the transplantation procedure is generally only performed in patients over 6 kg or six-months old. Pediatricians are therefore tasked with preventing the affected babies from hyperammonemia until they are old enough to undergo transplantation surgery. In these cases, hepatocyte transplantation offers a novel treatment to maintain low blood ammonia levels in patients not suitable for liver transplantation.

Clinical cell transplantation has been hampered by low engraft efficiencies, as well as a lack of real-time rejection diagnosis [9,10]. There is currently a need for research into cell transplantation and the development of pico-, or even femto-level liquid biopsies. In terms of donor cell preconditioning, cell revitalization improves engraftment since cells tend to be damaged by isolation, culturing, and storage procedures. The mitogen-activated protein kinase 4 (MKK4) inhibitor, myricetin (3,3′,4′,5,5′,7-hexahydroxylflavone) has this potential in hepatocytes. Myricetin is a natural flavonol that is widely found in plants and fruits including grapes, berries, and herbs, as well as vegetables and nuts [11]. Myricetin is known to exert antioxidative cytoprotective effects in various cells including a hepatoblast cell line [12,13,14,15]. The effects are thought to be mediated by the inhibition of mitogen-activated protein kinase kinase 4 (MKK4) and c-Jun N-terminal kinase (JNK) activation [16]. However, there are no reports on the application of myricetin in the culture of primary hepatocytes, which could increase the efficacy of cell transplantation. In this study, myricetin treatment was found to enhance cryopreserved porcine and human hepatocytes in vitro, thus improving the engraftment outcome compared to non-treated cells. Cell transplantation was found to prolong the survival periods of OTC-deficient Severe combined immunodeficiency (SCID) mice. Our findings demonstrate that myricetin could be used as a preconditioning treatment in order to improve the success rate of cell transplantation therapies.

## 2. Results

### 2.1. Hepatocyte Culture

#### 2.1.1. Porcine Hepatocyte Culture with Myricetin

Cryopreserved hepatocytes of a neonatal kusabira orange transgenic pig were cultured and showed the typical morphology of hepatic parenchymal cell with myricetin (Figure 1). Since kusabira orange expression is regulated by the albumin promoter gene, fluorescence intensity was used to indicate the existence of differentiated hepatic parenchymal cells (Appendix A). Compared to the control (Ctrl) and vehicle (DMSO) wells, cells treated with 1 µmol/L and 3 µmol/L of myricetin retained kusabira fluorescence until day 21 of culturing. In contrast, cells treated with 10 and 30 µmol/L of myricetin were exceptional on day 5 but the florescence intensity was as low as the control on days 10 and 21, suggesting that there is an optimum concentration of myricetin.

#### 2.1.2. Human Hepatocyte Culture with Myricetin

Cryopreserved human hepatocytes (lot. EJW and FLO) were cultured with 1–30 µmol/L myricetin (Figure 2a–j). The phase-contrast micrographs on culture day 7 showed the cells treated with 1 and 3 µmol/L of myricetin in both lots. Similar to porcine hepatocytes, human cells cultured at higher concentrations (10 and 30 µmol/L of myricetin) appeared to be attenuated, and the cell layer looked sparse. In terms of the functions of hepatic differentiation, nine typical genes were assayed (Figure 2k–s). Hepatocytes treated with 1 µmol/L and 3 µmol/L of myricetin showed a significant increase in the expression of all genes, including carbamoylphosphate synthetase I (r) and ornithine transcarbamylase (s), the key enzymes in the ammonia metabolizing urea cycle. However, a higher concentration of myricetin had no effect on gene expression except for tyrosine aminotransferase (n) and tryptophan 2,3-dioxygenase (o).

#### 2.1.3. Ammonia Removal Activity of Myricetin Treated Hepatocytes

The ammonia removal activity of kusabira orange porcine hepatocytes and human hepatocytes (EJW) is summarized in Figure 3. In porcine hepatocytes, the activity of the control and vehicle control (DMSO) was −56.7 ± 33.6 and −26.3 fmol/h/cell, respectively, indicating that the cells produced ammonia rather than being metabolized (Figure 3a). However, myricetin-treated hepatocytes were found to have decreased levels of ammonia. The activity of human hepatocytes increased significantly after treatment with 1–10 µmol/L of myricetin (Figure 3b). The results indicated that myricetin supports hepatocyte functions not only in terms of the levels of gene expression, but also in terms of the functions.

### 2.2. Hepatocyte Transplantation into Mouse Liver

#### 2.2.1. Fate of Hepatocytes Injected Directly into the Liver of Neonatal SCID Mice

Kusabira orange fluorescent pig hepatocytes had a good survival rate at six weeks (Figure 4a,b) and 15 weeks (Figure 4c,d) after transplantation into the livers of SCID mice, although the route of injection varied from the clinical procedure. The transplanted cells formed clusters, suggesting that the xenogeneic hepatocytes survived and colonized.

#### 2.2.2. Survival of Human Hepatocytes in SCID Mouse Livers and Semi-Quantitative Estimation of the Advantage of Myricetin Pretreatment

Human albumin immunohistochemistry revealed that myricetin-treated human hepatocytes (EJW) survived in the livers of SCID mice (Figure 5a–h); Figure 5a,b represent the control liver (injected with saline). The colonies of myricetin-treated human hepatocytes grew gradually in size at 3, 6, and 10 weeks after transplantation (Figure 5c–h). In contrast, the survival of non-treated hepatocytes was not marked in the liver 10 weeks after the transplantation (Figure 5i,j). To estimate the amount of transplanted xenogeneic hepatocytes, the levels of human albumin in the sera of SCID mice were determined (Figure 5k). The medians of the levels in mice transplanted with myricetin-treated and non-treated human hepatocytes were 21.9 and 1.93 ng/mL, respectively (*n* = 5 each; *p* < 0.05 by Wilcoxon rank sum test), suggesting that the myricetin-treated hepatocytes had a higher rate of survival than the non-treated hepatocytes.

#### 2.2.3. Therapeutic Potential of Myricetin Treated Human Hepatocytes in Mutant Mice

The survival times of ornithine transcarbamylase-deficient (OTCD) SCID mice were estimated after being transplanted with myricetin-treated and non-treated human hepatocytes (Figure 6a). While the median survival time of non-transplanted control mice was 32 days (range, 20–47 days; *n* = 5), the survival time of mice transplanted with non-treated and myricetin-treated hepatocytes was prolonged to 47 days (range, 20–52 days; *n* = 5) and 54 days (range, 52–69 days; *n* = 5), respectively. The difference between the control and myricetin-treated hepatocyte groups was found to be statistically significant by the Wilcoxon rank sum test. Figure 6b shows the appearance of changes in OTCD SCID mice 21 days after hepatocyte transplantation. The OTCD mice were characterized by sparse fur, whereas the cell-transplanted mice had a dense coat of fur, as well as an increase of body weight. As a result, the cell-transplanted mice had a longer survival time. Human albumin immunostaining of the autopsied livers also supported the prolonged survival of mice with transplanted cells.

### 2.3. Effect of Myricetin on the Phosphorylation of MKK4

Western blot analysis was performed for MKK4 and phosphorylated MKK4 (MKK4^p^) of cultured human hepatocytes (EJW and FLO) (Figure 7). MKK4^p^ of hepatocytes harvested at 5 or 10 min after the addition of 3 µmol/L of myricetin became faint (Figure 7a). Quantification also showed the reduction of MKK4^p^ 5 min (EJW) or 10 min (FLO) after the incubation, suggesting the inhibitory effect of myricetin on c-Jun N-terminal kinase (JNK)-dependent signaling pathway.

## 3. Discussion

The cytoprotective activity of myricetin is assumed to be a result of the inhibition of JNK-dependent signaling, which regulates cell proliferation and survival [15,16]. MKK4 and MKK7 are the key activators of JNK signaling [17], wherein the reciprocal regulation of MKK4 silencing and MKK7 induction are associated with hepatocyte regeneration in the murine model [18]. In addition, myricetin has anti-apoptotic properties via the regulation of PI3K/Akt and MAPK signaling pathways [15]. Thus, we performed the myricetin-treated hepatocyte transplantation.

It is worth noting the use of novel cell material, namely, the primary hepatocytes of albumin promotor-controlled kusabira orange transgenic pig in this study. Furthermore, we used hepatocytes of newborn pigs within 2 weeks after birth, which have high levels of proliferative activity and resistance to cryopreservation. In previous studies, we used kusabira orange transgenic pigs in which the expression was regulated by the cytomegalovirus (CMV) promotor [19,20], wherein, even if the hepatocytes showed fluorescence, it was uncertain whether the cells were in fact hepatic parenchymal cells or whether the cells maintained hepatic differentiated functions. When hepatocytes are cultured in vitro or transplanted into animals, the differentiated functions are easy to miss if cells are morphologically similar. Alternatively, contaminated non-parenchymal cells such as fibroblasts, endothelial cells, and cholangiocytes may become predominant and replace hepatic parenchymal cells. In the present study, since the expression of kusabira orange was regulated by the albumin promoter, fluorescence indicated the presence of functioning hepatic parenchymal cells [21]. In addition, due to the permeability advantage, engrafted cells beneath the surface were detectable in situ. As such, well-proliferated colonies of porcine hepatocytes were found in the livers of the transplanted mice.

Human hepatocytes of two different lots, EJW and FLO showed the effects of myricetin supplementation in both their morphological and functional properties. Similar to porcine hepatocytes, the optimum concentration ranged from 1 to 3 µmol/L in order to maintain the differentiated functions. The optimum concentration range coincides with a previous report that myricetin-treatment protected carcinogen-induced DNA damage in human hepatoma cell lines [14]. The higher concentration may cause cellular damage such as apoptosis. The expression levels of two typical urea cycle enzymes, carbamoylphosphate synthetase I and OTC were increased and, in support of this, the ammonia removal activity of human and porcine hepatocytes was also high.

In the cell transplantation experiment, myricetin-treated human hepatocytes were found to form large colonies, wherein the levels of human albumin in the sera of recipient mice were higher than those in the mice that received non-treated human hepatocytes. Cytological potency corresponded with the prolongation of survival of OTCD SCID mice.

To promote clinical cell transplantation, it will be crucial to improve cell viability after cryopreservation. Most types of cells are impaired by the process of freezing and thawing. Indeed, clinical islet transplantation is performed with freshly isolated islets and recent meta-analysis reported the inferiority of cryopreserved cells in the therapeutic efficacy of cell therapy for ischemic stroke [22]. Hepatocytes also easily lose their viability after freezing and thawing [23]. Recently, novel methods for the cryopreservation of hepatocytes have been reported to improve the viability after freezing and thawing, i.e., alginate gel encapsulation [24,25], and vitrification [26]. Surprisingly, the combination of cryoprotective supplements such as anti-apoptotic pan-caspase inhibitor, iron chelator, and human serum albumin and alginate gel encapsulation increased albumin production of cryopreserved hepatocytes rather than freshly isolated hepatocytes [24]. The cryopreservation of a large mass of encapsulated hepatocytes has also been reported [25]. Since alginate encapsulation has been found to enhance therapeutic efficacy [27], this method may accelerate the realization of hepatic cellular medicine. Vitrification, which is an effective technique for the cryopreservation of gametes, embryonic stem cells, induced pluripotent stem cells, etc., has been used on a small scale. Recently, successful large-scale cryopreservation of hepatocytes was reported as the bulk droplet vitrification method [26]. The method improved urea production and albumin synthesis compared with conventional cryopreservation methods. Neonatal hepatocytes have been proposed as an alternative source of hepatocytes, because they have excellent cell viability and functions after cryopreservation [28]. Our present method of myricetin pretreatment may support these novel technologies.

Also, myricetin has been found to exert rosiglitazone-like actions, acting as an accelerator of glucose utilization by ameliorating the impaired insulin-signaling pathway [29,30]. In other studies, treatment of rats with myricetin was found to improve not only streptozotocin-induced type I diabetes, but also insulin-resistant hyperglycemia induced by a high fructose diet [31]. Although myricetin has been found to reduce hyperglycemia in diabetic rats, possibly via its ability to increase hepatic glycogen synthesis and to normalize hypertriglyceridemia, insufficient information is currently available. One recent study reported that myricetin’s effect is due to direct inhibition of MEK, JAK1, Akt and MKK4 kinase activity [15]. Another study found that the anti-oxidation properties of myricetin are mediated by the suppression of the MKK4 and JNK signaling pathways [16]. In addition, MKK4 has been reported to be a key regulator of liver regeneration [18]. Thus, myricetin may have not only anti-oxidative but also regenerative effects. As such, we should consider the tumorigenesis of myricetin-treated cells because the inactivation of JNK signaling pathways may disrupt the normal balance of cell growth, especially under pathological stress. It will be necessary to investigate safety from the point of view of growth regulation.

In this study, we investigated the maintenance effect of myricetin against human hepatocyte functional capacity and its involvement in the MAPK pathways. Myricetin showed antioxidative, cytoprotective, anti-carcinogenic, antimicrobial, and anti-platelet properties [12,15,32,33,34], which are all beneficial for cell engraftment. We hope the results presented here will help to demonstrate that myricetin treatment has the potential to make a significant contribution to the development of cell transplantation therapies.

## 4. Materials and Methods

### 4.1. Animals

Kusabira orange transgenic pigs were established and bred in the Meiji University School of Agriculture using albumin promoter-containing kusabira-orange plasmid (Appendix A [19,21]). Immune deficient CB17 SCID mice (C.B-17/IcrHsd-Prkdc^scid^) were purchased from Japan SLC, Inc. OTCD carrier female sparse-fur (spf/1) mice (ID156, Balb/c back ground) were purchased from the Center for Animal Resources and Development at the Institute of Resource Development and Analysis, Kumamoto University. OTCD female mice were mated repeatedly with male CB17 SCID mice to establish OTCD-SCID mice in our institutional animal center. The genotypes of OTCD and SCID determinants were confirmed by PCR analysis from DNA obtained via a tail biopsy specimen (Appendix A [35]).

### 4.2. Cells and Chemicals

Porcine hepatocytes were isolated from a neonatal (<14 day-old) kusabira orange pig and cryopreserved as previously described [23]. Two lots of cryopreserved human hepatocytes, Lot. EJW (29-year-old Caucasian) and FLO (14-month-old Hispanic), were purchased from Celsis (In Vitro Technologies, Baltimore, MD, USA). Myricetin was purchased from Sigma-Aldrich (M6760, Sigma-Aldrich, St. Louis, MO, USA). All other chemicals were of reagent special grade.

### 4.3. Hepatocyte Culture

Cryopreserved hepatocytes (total volume, 1 mL) were thawed in a 37 °C water bath, dispersed in 10× volume of William’s E medium (W1878; Sigma) and centrifuged at 50× *g* for 1 min. The precipitated cells were resuspended in the medium and 1 × 10^6^ hepatocytes were cultured in 6-well collagen-coated plates (2352227; Becton Dickinson, Franklin Lakes, NJ, USA) with 2 mL of William’s E medium containing 1 µmol/L dexamethasone, 1 µmol/L insulin, 2 mmol/L GlutaMax (Gibco), 10% fetal bovine serum, and antibiotics (100 mg/mL penicillin G, 100 µg/mL streptomycin, 100 µg/mL kanamycin, and 250 ng/mL amphotericin B; FUJIFILM Wako Pure Chemical Corp., Osaka, Japan). Three hours after seeding, the appropriate amount of myricetin stock solution (20 mmol/L in dimethylsulfoxide) was added to the culture medium at a final concentration of 1–30 µmol/L. In the western blotting experiment, after the first 3 h culture, the medium was replaced with supplement-free medium for 24 h. Then, 3 µmol/L of myricetin was added, followed by harvesting of the hepatocytes at 5–30 min after. Ammonia removal activities were determined according to the method previously described [23]. Quantification of kusabira fluorescence was performed with ImageJ software (National Institute of Health, Bethesda, MD, USA) [36].

### 4.4. Hepatocyte Transplantation

The hepatocytes cultured with or without myricetin for 24 h (see above) were harvested with 0.05% trypsin-EDTA solution (25300120; Thermo Fisher Scientific, Waltham, MA, USA) and suspended at a concentration of 1 × 10^4^ cells/µL in saline. Then, 20 µL of cell suspension (2 × 10^5^ hepatocytes) was injected directly via the ventral surface into the livers of 2-day-old neonatal SCID or OTCD-SCID mouse using a micro-syringe (#705; Hamilton, Franklin, MA, USA) with a disposable 30G needle (#90130; Hamilton). The procedure was performed under anesthesia induced by intraperitoneal injection of a mixture of medetomidine hydrochloride, midazolam, and butorphanol tartrate.

### 4.5. Quantitative PCR Analysis

Total RNA (2 µg) was reverse-transcribed with oligo (dT), as described previously [37]. The glyceraldehyde-3-phosphate dehydrogenase PCR primers were used as a positive control for the human cDNAs. For the quantitative analysis of the mRNA levels of albumin, CYP3A4, alpha1-antitrypsin, tyrosine aminotransferase, tryptophan 2,3-dioxygenase, hepatocyte nuclear factor 4 alpha, cytokeratin 18, carbamoylphosphate syntherase I, ornithine transcarbamylase, the total RNA was isolated from 0 to 30 µmol/L of myricetin-treated human hepatocytes using an RNeasy mini-kit (Qiagen, Chatsworth, CA, USA) and reverse-transcribed by TaKaRa recombinant Taq (Takara Bio Inc., Shiga, Japan). The primer sets used are provided in Appendix A. Real-time PCR was carried out with an ABI PRISM 7000 Sequence Detection System. The 25 µL of reaction mixture contained 12.5 µL of SYBR Green PCR Master Mix (TOYOBO, Osaka, Japan), 10 ng of cDNA template and a primer set. The relative quantification of the transcripts was analyzed using the comparative threshold cycle method, according to the manufacturer’s instructions.

### 4.6. Immunohistochemical Analyses

Slices of the liver tissue sections (paraffin-embedded) were incubated with rabbit anti-human albumin antibody (Inter-Cell Technologies, Inc. Jupiter, FL, USA) overnight at 4 °C, followed by incubation with horseradish peroxidase (HRP)-conjugated secondary antibody (Histofine simple Stain, Nichirei Bioscience Inc, Tokyo, Japan). Staining was detected by diaminobenzidine and H_2_O_2_ (Histofine simple Stain). In Appendix A, cultured hepatocytes were fixed with 80% ethanol and stained with rabbit anti-porcine albumin antibody (ab79960, Abcam plc, Cambridge, UK), followed by incubation with fluorescence isothiocyanate (FITC)-conjugated secondary antibody (G0452, Tokyo Chemical Industry Co., Ltd. Tokyo, Japan).

### 4.7. Human Serum Albumin Assays in the Mouse Plasma

Mice sera were collected for the human serum albumin assays by Human Albumin ELISA Quantitation Set (Bethyl Laboratory, Montgomery, TX, USA). ELISA was then performed, according to the manufacturer’s instructions. Normal human and non-transplanted SCID mouse sera were tested in each experiment in addition to the standards included in the kit.

### 4.8. Western Blotting

Western blotting was performed as described previously [37]. Briefly, the cells were lysed in RIPA buffer (Santa Cruz Biotechnology, Inc., Santa Cruz, CA, USA) prior to use, followed by the addition of protease inhibitor and phosphatase inhibitor cocktail (#04080-11 and #06863-01; NACALAI TESQUE, INC., Kyoto, Japan). Blots were incubated with anti-MKK4 and anti-phospho-MKK4 (Ser257/Thr261) antibody (#9152 and #9156; Cell Signaling Technology, Danvers, MA, USA) overnight at 4 °C. After washing, the blots were incubated with HRP-conjugated secondary antibody (0.04 µg/mL) for 30 min. The blots were developed using an enhanced chemiluminescence substrate, according to the manufacturer’s instructions. Quantification was performed with ImageJ software [36].

### 4.9. Ethical Considerations

All animal experiments were approved (date of approval: 1 April 2013) by the Laboratory Animal Ethics Committee of the National Center for Child Health and Development (IRB number: A2000-001) based on the Japanese Guideline for Animal Experiments of Ministry of Health Labor and Welfare.

### 4.10. Statistical Analysis

Statistically significant results were determined using one-way ANOVA and Fisher’s protected least significant difference (PLSD) test (Figure 2 and Figure 3), or the Wilcoxon rank sum test (Figure 5 and Figure 6) by JMP11 (SAS Institute, Cary, NC, USA). A *p*-value < 0.05 was considered to be significant.

## 5. Conclusions

Myricetin treatment was effective in enhancing the survival of hepatocytes and the treatment lead to a better outcome in intrahepatic hepatocyte transplantation.

## Figures and Tables

**Figure 1 ijms-20-06123-f001:**
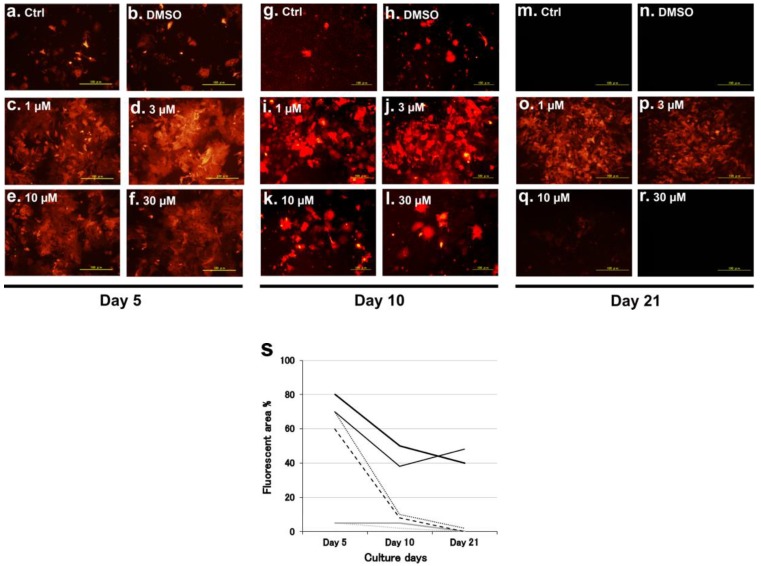
Fluorescent micrographs of the changes over time in the primary cultures of cryopreserved hepatocytes from a kusabira orange transgenic pig. Hepatocytes were cultured with 1–30 µmol/L of myricetin as indicated in each photo. Ctrl and DMSO denote the non-treated control and the vehicle (dimethyl sulfoxide) only, respectively. Scale bar: 100 µm. (**a**–**f**) Culture day 5, (**g**–**l**) culture day 10, (**m**–**r**) culture day 21. (**s**) Quantification of fluorescence positive areas. Ctrl; gray dotted line, DMSO; gray solid line, myricetin 1 µmol/L; black solid line, 3 µmol/L; black bold solid line, 10 µmol/L; black dotted line, 30 µmol/L; black broken line.

**Figure 2 ijms-20-06123-f002:**
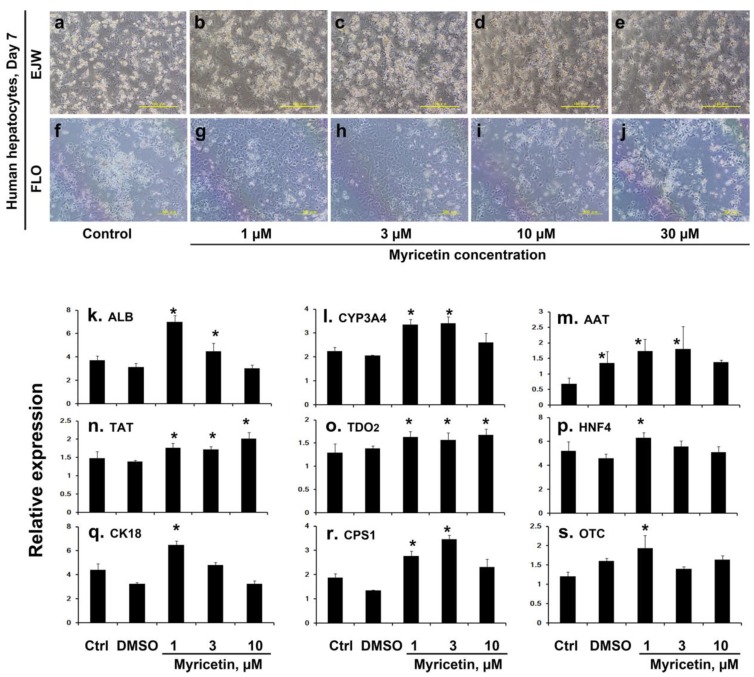
Phase-contrast micrographs and gene expression profiles of human cryopreserved hepatocytes 7 days after the start of culture. Hepatocytes were cultured with 1–30 µmol/L (10 µmol/L in quantitative PCR experiment) of myricetin as indicated in the lower margins. (**a**–**e**) The morphologies of human hepatocytes, lot. FLO. Scale bar: 500 µm (**f**–**j**) those of lot. EJW. Scale bar: 100 µm. (**k**–**s**) Quantitative PCR profiles of parenchymal hepatocyte specific genes. Samples were taken from EJW at culture day 7. ALB, albumin; CYP3A4, cytochrome P450 3A4; AAT, alpha1-antitrypsin; TAT, tyrosine aminotransferase; TDO2, tryptophan 2,3-dioxygenase; HNF4, hepatocyte nuclear factor-4 alpha; CK18, cytokeratin 18; CPS1, carbamoylphosphate synthetase I; OTC, ornithine transcarbamylase. The primers used are described in Appendix A. Asterisks indicate statistically significant results (* *p* < 0.05) compared to controls using one-way ANOVA and Fisher’s protected least significant difference (PLSD) test.

**Figure 3 ijms-20-06123-f003:**
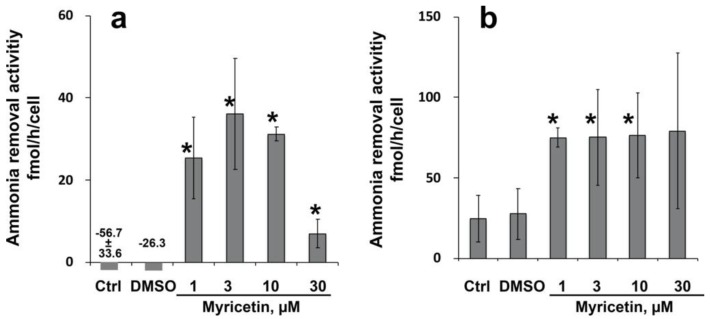
Ammonia removal activity of cryopreserved porcine and human hepatocytes cultured with 1–30 µmol/L of myricetin for 7 days. (**a**) Ammonia removal activity of kusabira orange transduced hepatocytes. The activity of control (Ctrl) and vehicle control (DMSO) were −56.7 ± 33.6 and −26.3 fmol/h/cell, respectively. (**b**) Ammonia removal activity of human hepatocytes (EJW). Asterisks (*) indicate statistically significant results compared to the controls using one-way ANOVA and Fisher’s protected least significant difference (PLSD) test.

**Figure 4 ijms-20-06123-f004:**
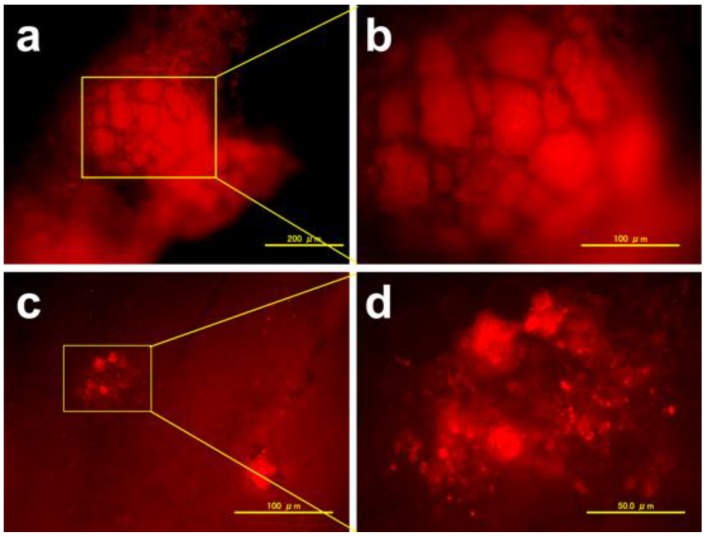
Fluorescent micrographs of Severe combined immunodeficiency (SCID) mouse liver after transplantation of kusabira orange transgenic pig hepatocytes. Hepatocytes were cultured with 3 µm/mol of myricetin for 24 h and harvested before transplantation. Cell suspensions (2 × 10^5^ cells in 20 µL saline) were injected directly via ventral surface of 2 days old SCID mouse livers using a micro-syringe. (**a**,**b**) 6 weeks after transplantation. Scale bar: 200 µm in (**a**) and 100 µm in (**b**). (**c**,**d**) 15 weeks after transplantation. Scale bar: 100 µm in (**c**) and 50 µm in (**d**).

**Figure 5 ijms-20-06123-f005:**
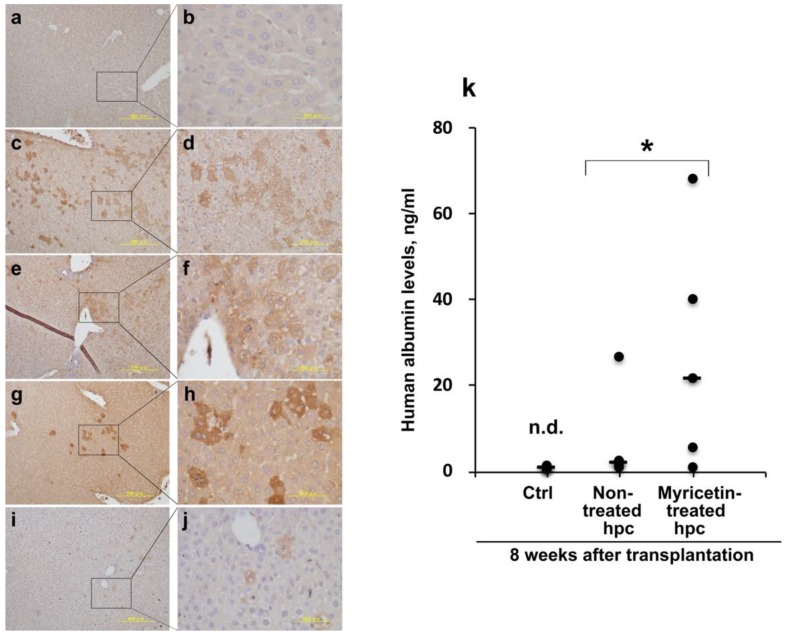
Human albumin immunohistochemistry and human albumin determination after transplantation of human hepatocytes (EJW) into SCID mouse liver. Hepatocytes were cultured with 3 µmol/L of myricetin for 24 h and harvested. The cell suspension (2 × 10^5^ cells in 20 µL saline) was injected directly via the ventral surface of 2 days old SCID mouse livers using a micro-syringe. (**a**,**b**) normal SCID mouse liver (control). (**c**,**d**) Liver 3 weeks after transplantation of myricetin-treated hepatocytes. (**e**,**f**) Liver 6 weeks after transplantation of the hepatocytes. (**g**,**h**) Liver 10 weeks after transplantation of the hepatocytes. (**i**,**j**) Liver 10 weeks after transplantation of non-treated human hepatocytes (EJW). Scale bars: 200 µm in (**a**), (**c**), (**e**), (**g**), (**i**) and 50 µm in (**b**), (**d**), (**f**), (**h**), (**j**). (**k**) human albumin levels in SCID mice sera 8 weeks after transplantation (closed circles, *n* = 5 each). Control (Ctrl) mice were injected with saline alone, wherein human albumin was not detectable (n.d.). The medians of the human albumin levels in the groups of mice transplanted with non-treated and myricetin-treated hepatocytes were 1.93 and 21.9 ng/mL (bars), respectively. Asterisks (*) indicate statistically significant results between the two experimental groups, calculated using Wilcoxson rank sum test.

**Figure 6 ijms-20-06123-f006:**
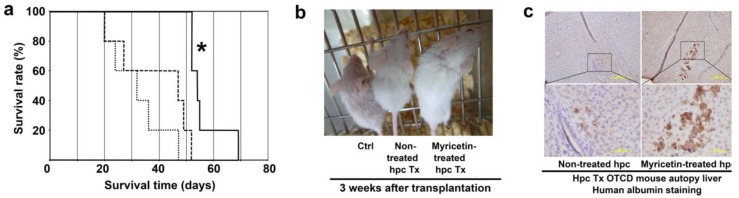
Survival time, appearance change, and autopsy after transplantation of human hepatocytes (EJW) into ornithine transcarbamylase-deficient (OTCD) SCID mouse livers. Hepatocytes were cultured with or without 3 µmol/L of myricetin for 24 h and harvested. The cell suspension (2 × 10^5^ cells in 20 µL saline) was injected directly via the ventral surface of 2 days old OTCD SCID mouse livers using a micro-syringe. (**a**) Survival time (day) of OTCD mice after transplantation of non-treated (broken line, *n* = 5; median, 47; range, 20–52) and myricetin-treated (solid line, *n* = 5; median, 54; range, 52–69) hepatocytes. Control mice received saline only (dotted line, *n* = 5; median, 32; range, 20–47). The day previous to the day that mice were found dead was denoted as the end of survival day. Asterisks indicate statistically significant results (* *p* < 0.05) between the control mice and the mice transplanted with myricetin-treated hepatocytes, using the Wilcoxon rank sum test. (**b**) Changes in the appearance of OTCD SCID mice 21 days after hepatocyte transplantation. Left, control mouse; middle, mouse transplanted with non-treated hepatocytes; right; mouse transplanted with myricetin treated hepatocytes. The cell-transplanted mice appeared to be healthy in a stepwise fashion, in terms of weight gain and hair growth. (**c**) Human albumin immunostaining of autopsied liver of mice transplanted with non-treated hepatocytes (left) and myricetin treated hepatocytes (right). Scale bars: 200 µm in upper left and right and 50 µm in lower left and right.

**Figure 7 ijms-20-06123-f007:**
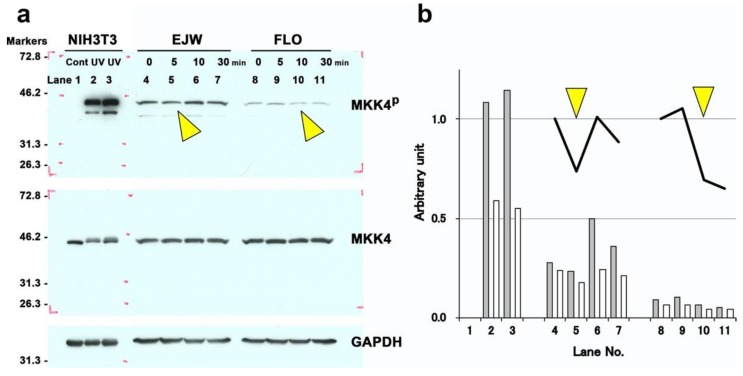
(**a**) Western blot results of mitogen-activated protein kinase 4 (MKK4) and phosphorylated MKK4 (MKK4^p^) of cultured human hepatocytes (EJW and FLO). Hepatocytes were harvested 5–30 min after the addition of 3 µmol/L of myricetin. Ultraviolet-irradiated NIH3T3 fibroblasts were used as the positive control for MKK4 phosphorylation. The arrowheads denote a reduction in phosphorylation. Markers, molecular weight markers. (**b**) Quantification of western blot. The values obtained by the image analysis were standardized by the intensities of Glyceraldehyde-3-phosphate dehydrogenase (GAPDH). Shaded bars and open bars indicate the intensities of MKK4^p^ and the proportions of MKK4^p^/total MKK4 (i.e., MKK4^p^ + MKK4). Solid lines indicate the time course changes of the proportion started with 1.0. The yellow triangles denote the reduction in phosphorylation.

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
