# Peer review of "Natural Flavonol, Myricetin, Enhances the Function and Survival of Cryopreserved Hepatocytes In Vitro and In Vivo"

_ijms, 2019, doi:10.3390/ijms20246123_

Round 1

Reviewer 1 Report

The manuscript by Cui et al makes novel discoveries regarding unique beneficial potential for the function and survival of cryopreserved hepatocytes.  Though there is accumulating the evidence regarding the therapeutic potential of hepatocyte transplantation in congenital metabolic diseases, it remains unclear how to improve the efficacy.  Authors showed that the treatment with MKK4 inhibitor, myricetin enhanced the function and survival of hepatocytes in vitro.  The transplantation of myricetin treated human hepatocytes revealed the high levels of human albumin in the sera of SCID mice compared to the control (non treated-hepatocytes).  They also showed the transplantation of myricetin treated hepatocytes prolonged the survival of ornithine transcarbamylase-deficient SCID mice. 

The studies are elegant, carefully designed with use of particular both porcine (kusabira orange expression is regulated by the albumin promoter gene) and human hepatocytes.

Minor comment

There seems to be an optimum concentration of myricetin.  How is the effect of myricetin to the apoptosis when hepatocytes are treated with high concentration of myricetin.

In Figure 6b, the panel should be replaced, since it is unclear that the

changes in the appearance OTCD SCID mice after transplantation.

Authors should discuss the risk of developing malignancy from transplanted hepatocytes.

Author Response

We appreciate reviewers sincerely for detailed comments. We revised our manuscript according to the comments

Comments by reviewer 1

The manuscript by Cui et al makes novel discoveries regarding unique beneficial potential for the function and survival of cryopreserved hepatocytes. Though there is accumulating the evidence regarding the therapeutic potential of hepatocyte transplantation in congenital metabolic diseases, it remains unclear how to improve the efficacy. Authors showed that the treatment with MKK4 inhibitor, myricetin enhanced the function and survival of hepatocytes in vitro. The transplantation of myricetin treated human hepatocytes revealed the high levels of human albumin in the sera of SCID mice compared to the control (non treated-hepatocytes). They also showed the transplantation of myricetin treated hepatocytes prolonged the survival of ornithine transcarbamylase-deficient SCID mice. The studies are elegant, carefully designed with use of particular both porcine (kusabira orange expression is regulated by the albumin promoter gene) and human hepatocytes.

Minor comment

There seems to be an optimum concentration of myricetin. How is the effect of myricetin to the apoptosis when hepatocytes are treated with high concentration of myricetin. In Figure 6b, the panel should be replaced, since it is unclear that the changes in the appearance OTCD SCID mice after transplantation. Authors should discuss the risk of developing malignancy from transplanted hepatocytes.

Reply to reviewer 1

We added consideration about the effect of higher concentration of myricetin in Discussion (line 216-218) We replaced the photo to specify the improvement of mice clearly. We added the risk of tumorigenesis of transplanted hepatocytes which were pretreated with myricetin in Discussion (line 235-240)

Reviewer 2 Report

The authors reported the positive effects of Myricetin in cryopreserved hepatocytes using both in vitro and in vivo experiments.

Abstract: lines 16: Explain the abbreviation MKK4; line 17: “cultured well”, not clear.

Introduction: Lines 61: the authors should add the role of MKK and the control of oxidative stress. Line 66: “cell therapy”: too vague.

Results: Line 75: “good appearance”; line 92 “with the cell layer looking to be loose”; line 195 “infantile”: use only scientific style.

Line 77-79: explain this sentence and add reference:  “Since kusabira orange expression is regulated by the albumin promoter gene, fluorescence intensity was used to indicate the existence of differentiated hepatic parenchymal cells”.

Figure 1: a quantification of fluorescence with appropriated statistical analysis should be added.

Line 89: “liveliness” not clear.

Line 90-91: Add: “human cell cultured…”.

Figure 3: the DMSO group is absent in figure 3A;

Line 124: add 2.2.1 “Fate of pig hepatocytes….”

The authors reported the use of hepatocytes cultured with or without myricetin for 24 h in the method section but in the figure 4 the analysis of liver after transplantation with pig hepatocytes without myricetin is absent. The same occurs for human hepatocytes: (Figure 5 a-b: control liver, injected with saline;  colonies of myricetin-treated human hepatocytes Figure 5 c-h). Add the results obtained with hepatocytes cultured without myricetin.

Figure 5 I: bars of medians: verify the values reported in the text.

Figure 7: a quantification of WB results with appropriated statistical analysis should be added. 

Discussion

The discussion should be improved  comparing the present protocol of hepatocyte preservation with those recently published.

Explain why low concentrations of Myricetin are better than high concentrations

Line 193: “Thus, we performed the myricetin-treated hepatocyte transplantation”: unclear

Eliminate “(Figure..)” from the discussion section.

Methods:

Add the genotypes of OTCD and SCID.

Author Response

We appreciate reviewers sincerely for detailed comments. We revised our manuscript according to the comments.

Comments by reviewer 2

The authors reported the positive effects of Myricetin in cryopreserved hepatocytes using both in vitro and in vivo experiments.

Abstract: lines 16: Explain the abbreviation MKK4; line 17: “cultured well”, not clear.

Introduction: Lines 61: the authors should add the role of MKK and the control of oxidative stress. Line 66: “cell therapy”: too vague.

Results: Line 75: “good appearance”; line 92 “with the cell layer looking to be loose”; line 195 “infantile”: use only scientific style.

Line 77-79: explain this sentence and add reference:

“Since kusabira orange expression is regulated by the albumin promoter gene, fluorescence intensity was used to indicate the existence of differentiated hepatic parenchymal cells”.

Figure 1: a quantification of fluorescence with appropriated statistical analysis should be added.

Line 89: “liveliness” not clear.

Line 90-91: Add: “human cell cultured…”.

Figure 3: the DMSO group is absent in figure 3A;

Line 124: add 2.2.1 “Fate of pig hepatocytes….”

The authors reported the use of hepatocytes cultured with or without myricetin for 24 h in the method section but in the figure 4 the analysis of liver after transplantation with pig hepatocytes without myricetin is absent. The same occurs for human hepatocytes:

(Figure 5 a-b: control liver, injected with saline; colonies of myricetin-treated human hepatocytes Figure 5 c-h). Add the results obtained with hepatocytes cultured without myricetin.

Figure 5 I: bars of medians: verify the values reported in the text.

Figure 7: a quantification of WB results with appropriated statistical analysis should be added.

Discussion

The discussion should be improved comparing the present protocol of hepatocyte preservation with those recently published. Explain why low concentrations of Myricetin are better than high concentrations

Line 193: “Thus, we performed the myricetin-treated hepatocyte transplantation”: unclear

Eliminate “(Figure..)” from the discussion section.

Methods:

Add the genotypes of OTCD and SCID.

Reply to reviewer 2

[Abstract]

Lines 16: Explain the abbreviation MKK4;

REPLY: We added the explanation of MKK4 in line 16 and listed it in the section of abbreviations.

Line 17: “cultured well”, not clear;

REPLY: We changed the words “cultured well” to “cultured stable” in line 18.

[Introduction]

Lines 61: the authors should add the role of MKK and the control of oxidative stress;

REPLY: We added the description “The effects are thought to be mediated by the inhibition of MKK4 and JNK activation [16].” in line 65-66.

Line 66: “cell therapy”: too vague;

REPLY: We changed “cell therapy” to “cell transplantation” in line 67.

[Results]

Line 75: “good appearance”; line 92 “with the cell layer looking to be loose”; line 195 “infantile”: use only scientific style;

REPLY: We changed the words “good appearance”, “with the cell layer looking to be loose”, and “infantile” to “stable appearance”, “with the cell layer looking to be sparse”, and “we used hepatocytes of newborn pigs within 2 weeks after birth which” in lines 77, 90, and 197, respectively.

Line 77-79: explain this sentence and add reference: “Since kusabira orange expression is regulated by the albumin promoter gene, fluorescence intensity was used to indicate the existence of differentiated hepatic parenchymal cells”

REPLY: We added data in supplementary figure 1 which shows that only albumin-positive hepatic parenchymal cells had kusabira fluorescence but albumin-negative hepatic non-parenchymal cells had not. Also, we added reference 28.

Figure 1: a quantification of fluorescence with appropriated statistical analysis should be added.

REPLY: Quantification data was added in Figure 1. Unfortunately, we did not have enough data for statistical analysis.

Line 89: “liveliness” not clear.

REPLY: We omitted the word “liveliness” in line 87.

Line 90-91: Add: “human cell cultured…”.

REPLY: We added “human cell cultured…” in line 88.

Figure 3: the DMSO group is absent in figure 3A;

REPLY: We added the DMSO group data in Figure 3A.

Line 124: add 2.2.1 “Fate of pig hepatocytes….”The authors reported the use of hepatocytes cultured with or without myricetin for 24 h in the method section but in the figure 4 the analysis of liver after transplantation with pig hepatocytes without myricetin is absent. The same occurs for human hepatocytes: (Figure 5 a-b: control liver, injected with saline; colonies of myricetin-treated human hepatocytes Figure 5 c-h). Add the results obtained with hepatocytes cultured without myricetin.

REPLY: We added histology of SCID mouse liver 10 weeks after the transplantation of non-treated human hepatocytes (Figure 5 (i) and (j)).

Figure 5 I: bars of medians: verify the values reported in the text.

REPLY: We are very sorry for mistake and thank you very much for your careful reading. Data were corrected to non-treated; 1.93 ng/ml and myricetin-treated; 21.9 ng/ml in the text and figure legend.

Figure 7: a quantification of WB results with appropriated statistical analysis should be added.

REPLY: Quantification data was added in Figure 7. Unfortunately, we did not have enough data for statistical analysis.

[Discussion]

The discussion should be improved comparing the present protocol of hepatocyte preservation with those recently published. Explain why low concentrations of Myricetin are better than high concentrations

REPLY: We added consideration about the effect of higher concentration of myricetin in Discussion (line 216-218)

Eliminate “(Figure..)” from the discussion section.

REPLY: We eliminated them.

[Methods]

Add the genotypes of OTCD and SCID

REPLY: We added supplementary table 2 which describes the genotyping of OTCD SCID mouse.

Round 2

Reviewer 2 Report

Because for Figure 1 and Figure 7 the authors reported "..we did not have enough data for statistical analysis ", the comment of these results should be avoided in the discussion as well as in the conclusion section.
The authors should compare their results with the findings from other studies about protocols of cryopreserved hepatocytes.
The authors replaced “cultured well” with “cultured stable” as well as “good appearance” with ”stable appearance”. Probably using “well cultured” or “good appearance” the authors wanted to indicate a good cell morphology that is completely absent using "stable appearance".

Author Response

Thank you very much for your advices. We revised our manuscript according to your comments.

Comments from reviewer 2, 2nd
Because for Figure 1 and Figure 7 the authors reported "..we did not have enough data for statistical analysis ", the comment of these results should be avoided in the discussion as well as in the conclusion section.
The authors should compare their results with the findings from other studies about protocols of cryopreserved hepatocytes.
The authors replaced “cultured well” with “cultured stable” as well as “good appearance” with ”stable appearance”. Probably using “well cultured” or “good appearance” the authors wanted to indicate a good cell morphology that is completely absent using "stable appearance".

1. Because for Figure 1 and Figure 7 the authors reported "..we did not have enough data for statistical analysis ", the comment of these results should be avoided in the discussion as well as in the conclusion section.
REPLY: We omitted the description about the data of Figure 1 and Figure 7 in which statistical analyses were not performed. (line 208-210 and line 226-227 in Discussion and line 352 in Conclusion)
2. The authors should compare their results with the findings from other studies about protocols of cryopreserved hepatocytes.
REPLY: We added a paragraph about the comparison of our results and other studies of cryopreservation in Discussion (line 229-248)
3. The authors replaced “cultured well” with “cultured stable” as well as “good appearance” with ”stable appearance”. Probably using “well cultured” or “good appearance” the authors wanted to indicate a good cell morphology that is completely absent using "stable appearance".
REPLY: We changed the expression “stable” in Abstract and “in a stable appearance” in Results to “.. showing the typical morphology of hepatic parenchymal cell under 1-10 µmol/l of myricetin, keeping hepatocyte specific gene expression, and ammonia removal activity.’’ in Abstract (lane 18-19) and to “showing the typical morphology of hepatic parenchymal cell” in Results (lane 78).